# Evaluation of Knowledge, Attitudes and Practices Related to Self-Testing Procedure against COVID-19 among Greek Students: A Pilot Study

**DOI:** 10.3390/ijerph19084559

**Published:** 2022-04-10

**Authors:** Georgios Marinos, Dimitrios Lamprinos, Panagiotis Georgakopoulos, Evangelos Oikonomou, Georgios Zoumpoulis, Nikolaos Garmpis, Anna Garmpi, Eirini Tzalavara, Gerasimos Siasos, Georgios Rachiotis, Anastasia Papaioannou, Dimitrios Schizas, Christos Damaskos

**Affiliations:** 1Department of Hygiene, Epidemiology and Medical Statistics Medical School, National and Kapodistrian University of Athens, 75, M. Asias Str., 11527 Athens, Greece; gmarino@med.uoa.gr; 2Emergency Care Department, Laiko General Hospital, 11527 Athens, Greece; panos.k.georgakopoulos@gmail.com (P.G.); gzoumpoulis@yahoo.gr (G.Z.); renatzalavara@gmail.com (E.T.); 3First Department of Cardiology, Hippokration General Hospital, Medical School, University of Athens, 11527 Athens, Greece; boikono@gmail.com (E.O.); gsiasos@med.uoa.gr (G.S.); 4Third Department of Cardiology, Thoracic Diseases General Hospital Sotiria, Medical School, University of Athens, 11527 Athens, Greece; 5Second Department of Propedeutic Surgery, Laiko General Hospital, Medical School, National and Kapodistrian University of Athens, 11527 Athens, Greece; nikosg22@hotmail.com; 6N.S. Christeas Laboratory of Experimental Surgery and Surgical Research, Medical School, National and Kapodistrian University of Athens, 11527 Athens, Greece; x_damaskos@yahoo.gr; 7First Department of Propedeutic Internal Medicine, Laiko General Hospital, Medical School, National and Kapodistrian University of Athens, 11527 Athens, Greece; annagar@windowslive.com; 8Department of Hygiene and Epidemiology, Faculty of Medicine, University of Thessaly, 41500 Larissa, Greece; grachiotis@gmail.com; 9Health Center of Nea Makri, 19005 Attica, Greece; anpapai@yahoo.com; 10First Department of Surgery, Laikon General Hospital, National and Kapodistrian University of Athens, 11527 Athens, Greece; dschizas@med.uoa.gr; 11Renal Transplantation Unit, Laiko General Hospital, 11527 Athens, Greece

**Keywords:** COVID-19, SARS-CoV-2, students, self-testing, pandemic

## Abstract

The COVID-19 pandemic has had a major impact on health, economy, society and education. In the effort to return to normalcy, according to the instructions of the Greek Government for the resumption of the operation of schools, a screening Rapid Antigen Detection Test with the method of self-testing is required for students twice per week, for the early identification and isolation of positive cases. We aimed to pivotally investigate the knowledge, attitudes and practices related to self-testing procedures against COVID-19 among Greek students. A questionnaire was distributed to a convenient sample of students in the region of Athens. Information about the vaccination coverage against SARS-CoV-2 was also obtained. Our study included 1000 students, with 70% of them having an average grade at school. Most of the participants were aware of coronavirus (98.6%) and the self-test (95.5%). The vast majority of students (97%) performed self-testing twice per week, with the 70% them being assisted by someone else. Nearly one sixth of the participants had been infected by COVID-19 (14%) while 36% of them have already been vaccinated against SARS-CoV-2. In conclusion, we report high compliance with the COVID-19 self-testing procedure among students in Attica, Greece. Older age adolescents are more likely to not comply with the regulations of self-testing. Consequently, tailored interventions targeted at older age adolescents are warranted in order to increase the acceptability of self-testing.

## 1. Introduction

During 2019, the world faced a new health threat called Coronavirus disease or COVID-19, caused by the virus named SARS-CoV-2, an RNA virus of the family of Coronaviridae. The severe acute respiratory syndrome caused by Coronavirus 2 (SARS-CoV-2) and the Coronavirus disease 2019 (COVID-19) led the World Health Organization (WHO) to characterized it as a pandemic, on 11 March 2020 [1]. Over 416 million positive cases have been recorded around the world [2]. Governments all over the world had to take emergency measures to minimize the risk of infection and the spread of the virus. Several studies for influenza transmission have already concluded that school closure may reduce transmission during influenza outbreaks [3,4,5,6]. In accordance with this view, schools, universities and colleges closed, at a nationwide level, which in most countries interrupted the school year [7]. Greek authorities took the decision to also implement this measure and shift the educational process towards online classes to reduce social contact between students and therefore interrupt transmission. Even in the early stages of the pandemic, it was proposed that rapid point-of-care tests could be used as a screening tool and contribute to the early diagnosis and control of COVID-19 transmission [8]. Studies during the return to normalcy period have shown that organized self-testing programs were well accepted by the population and that asymptomatic testing could help in the establishment of low COVID-risk permitted activities [9,10]. Studies for regular asymptomatic screening tests in school environments have shown that this is a way to control COVID-19 transmission [11,12].

After the lockdown period, the Greek Government, in an attempt to reopen schools, adopted the use of Rapid Antigen Detection Tests (RADTs) to curtail and control local outbreaks of COVID-19. Each student is obliged to perform two weekly RADTs with the method of self-testing. Self-testing is defined as the process in which a person collects their own specimen from their nose/throat (nose swab, throat swab, saliva, or combination of those), proceeds to conduct the test and interprets the results themselves. The results have to be uploaded to an online platform created by the government. Each positive test has to be confirmed with another RADT or Polymerase Chain Reaction test (PCR test) by an authorized bearer. The purpose of this procedure is to protect Public Health by detecting asymptomatic carriers of the SARS-CoV-2 virus easily, quickly and reliably [13,14].

On May 2021, the Food and Drug Administration (FDA) and European Medicine Agency (EMA) approved the use of COVID-19 vaccines for persons over the age of 12 years old [15,16]. However, debate exists about the need for vaccination in adolescents, with supporters underlining its importance in the protection of the whole society [17]. A study from Wong WHS et al. during the first months of vaccine approval for teenagers recorded a medium acceptability, with 39% of the participants holding the intention of being vaccinated against COVID-19 [18]. In Greece, since July 2021, a COVID-19 vaccine has been added to the National Vaccination Program for children and adolescents aged 12–17 [19].

The definition of electronic bullying or Cyberbullying is derived from the traditional concept of bulling and refers to the use of information and communication technologies to support deliberate, repeated, and hostile behavior by an individual or group, that is intended to harm others, with major impacts on adolescent behavior and mental health [20,21]. To the best of our knowledge there are no studies reporting on the impact of electronic bullying on preventive measures against COVID-19.

The aim of this study was to evaluate the knowledge of Greek students regarding the self-testing procedure against COVID-19, to investigate the associated factors that contribute to the implementation and the management of the RADTs, as well as to find a correlation of the results between the students’ performance and the existence of electronic bullying in the school environment. The secondary aim was to pivotally investigate the vaccination coverage against COVID-19 among Greek Students.

## 2. Materials and Methods

### 2.1. Study Design and Sample

The study was conducted in a convenient sample of schools in Athens, with a total population of almost 4 million people. The study was approved by the Hospital Ethics Committee and was conducted according to the principles of the Declaration of Helsinki of 1975, as revised in 2008. Inclusion criteria were the voluntary involvement of students living in the region of Attika. Common exclusion criteria include the presence of a severe and uncorrectable cognitive, visual, or hearing impairment that would preclude a participant’s ability to complete the questionnaire. Participants were not required to record their personal details. Data were collected from 20 September to 20 October 2021. Each survey was estimated to take less than 10 min to complete. In total, 1000 students from 13 to 18 years old were selected and sub-divided into the following two groups: 13–15 years old and 16–18 years old (y.o.). A signed parent consent form was received prior to the completion of the questionnaire.

### 2.2. Questionnaire

Survey items were adapted from questionnaires used to study prior outbreaks [22,23]. The questionnaire consisted of 28 items designed to assess students’ knowledge of SARS-CoV-2, their knowledge, attitudes and practices on self-testing and to obtain information about the school environment. More specifically, the survey included questions such as “What is COVID-19?”, “Do you know what a self-test is?” and “Do you take a self-test twice per week”. The questionnaire also included questions regarding school grades and school bullying: “What is your average grade at school?”, “Does anyone bully you at school?” and “Have you ever bullied another student?”. The survey also included questions on demographics (sex and age) and COVID-19 vaccination coverage.

### 2.3. Statistical Analysis

Categorical variables were presented as absolute (N) and relative frequencies (percentages; %). Descriptive statistics were calculated for all participant characteristics and survey responses and were presented as valid percentages and as numerical values where appropriate. The chi square test (χ^2^) was used to determine statistically significant differences among subgroups of the study’s participants. Furthermore, 95% Confidence Intervals (C.I.) were calculated. A *p* value < 0.05 was considered as statistically significant. Lastly, SPSS 23 for Windows was used for the analysis.

## 3. Results

From the total group of 1000 students aged between 13 and 18 years, 50.3%were females. Most of the participants (98.6%) reported that they were informed of COVID-19, while 13.5% of them had been infected and 23.7% had a friend or family member who was positive during the pandemic. The overwhelming majority (97%) of the students reportedly performed self-testing twice per week and 74.8% were submitted to another test to diagnose infection; either PCR (31.1%) or RADTs (43.7%). Almost the same percentage (85.4%) of family members also performed a diagnostic test for COVID-19. Only 6.4% of the participants confirmed the result of the self-test with another test, either PCR (19.4%) or RADTs (80.6%). Among the teenagers, 76.1% reported that they performed the self-test with the help of one parent. Two thirds of the students (69.3%) faced a problem while performing the self-test. A total of 22.6% of them faced difficulties with the procedure of taking the sample and 37.5% of them had trouble submitting the results on the dedicated online platform. Additional problems were found while dispensing drops of the buffer-specimen mixture into the sample well on the coronavirus antigen test cassette (6.6%) and during the time waiting for the results (2.5%). A sufficient percentage (69.2%) of the students experienced fear during the procedure of self-testing, with half of them reported to be a bit (35.5%) or very scared (39%). Information about the disposal of the waste of the procedure were also obtained. More specifically, 72% and 74% of the responders answered correctly about the disposal of the negative and the positive tests, respectively. On the contrary, half of the students responded wrongly to the question as to whether it is correct to discard the self-tests in a recycling bin or the drainage. A total of 36% of the participants have been vaccinated against COVID-19, with most of them declaring that they fear any possible side effects of the vaccine. An interesting finding of our study is that almost one third of the students (30%) have been bullied at least one time at school and 28.1% have bullied their classmates at least once (Table 1). The univariate analysis demonstrates that the students who were 16–18 years old and not yet infected from COVID-19 use self-tests more often (*p* = 0.04), whilst younger teenagers (13–15 y.o.) seem to be more obedient with the regulations and they more often perform a RADT (*p* < 0.001). The positive family environment, school bullying, position as victims or perpetrators, and school performance are not significantly associated with the procedure of self-testing in both age categories (Table 2, Table 3 and Table 4).

## 4. Discussion

The data of our survey show that government instructions regarding self-testing procedure have been followed by an overwhelming majority of teenage students after the reopening of schools. While the number of positive cases in Greece is increasing and even surpassing 40.000 per day, the low recorded percentage of students infected from COVID-19 in our study shows that the use of self-testing RADTs could have been a vital part of helping the Greek government to limit the spread. According to the current data, over 40 million self-testing RADTs have been performed [24,25]. Another interesting point of our study is that over two thirds of the students were familiar with how to properly dispose of the medical wastes of tests, showing that the students are most probably familiar with ways of transmission, such as in the awareness that RADTs contain nasal droplets. In a recent survey of more than 1200 socio-demographically diverse teenagers, it was found that most of the students were informed about SARS-CoV-2 and more than half of them were aware of the ways of transmission [26]. The proper and safe disposal of medical waste is related to the prevention and control of the epidemic situation [27]. Hence, although self-testing has been ongoing since April 2021 [28], almost 70% of the teenagers stated that they felt fear during the procedure of the test. In order to address and reduce this sense of fear, it is critical for the Greek government to organize new educational campaigns, establishing continuity in the government measures and maintaining or even increasing the participation of the population. As almost 40% of the participants faced a problem during their online statement of the results, a new approach in reporting a result will need to be employed. As Goggolidou et al. mention in their study, specially designed applications or dedicated, easily accessible websites, could positively influence the success of the self-test procedure [29]. Older age adolescents are more likely to not comply with the regulations of self-testing, most probably due to the fact that as teenagers move on to puberty, they become more reactive when rules are imposed on them [30]. The vaccination coverage of the students is similar to the recorded data in the US [31]. More studies should be performed to determine the reasons behind the low rate of vaccination coverage among adolescents. The recorded bullying in our study is similar to the results of Vaillancourt et al., with a rate of 30% [32]. Being a victim or perpetrator of bullying seems not to have any effect on self-testing among the participants, which is a desired result as COVID-19 has become a cause of bullying and has stigmatized specific population groups [33]. Our study did not find a correlation between school performance or positive family environment with the implementation of RADTs. Our results are subject to several limitations which needs to be considered prior to the interpretation of the results. Firstly, the survey was conducted among a convenient sample of students only in the Attika region. However, our study includes girls and boys from socioeconomically diverse schools. Prior research on virus outbreaks guided the creation of the survey [22,23].

## 5. Conclusions

Herein, we report a high acceptability of self-testing among teenaged students and also a high compliance with the COVID-19 self-testing procedure among students in Attica, Greece. In addition, older age adolescents are more likely to not comply with the regulations of self-testing. Consequently, tailored interventions targeting older age adolescents are warranted in order to increase the acceptability of self-testing.

## Figures and Tables

**Table 1 ijerph-19-04559-t001:** Descriptive statistics of the answers divided into two age subgroups.

Questions	13–15 Year Old N (%)	95% CI [LL, UL]	16–18 Years Old N (%)	95% CI [LL, UL]
Sex				
Male	261 (52.2)	48–57	242 (48.4)	44–53
Female	239 (47.8)	43–52	258 (51.6)	47–56
Have you ever heard about COVID-19?				
Yes	488 (97.6)	96–98	498 (99.6)	98–99
No	12 (2.4)	1–4	2 (0.4)	0.1–2
Have you been infected from COVID-19?				
Yes	80 (16)	13–19	55 (11)	8–14
No	420 (84)	80–87	445 (89)	86–91
Does anyone of your friends/family was positive to COVID-19?				
Yes	170 (34)	30–38	67 (13.4)	11–18
No	330 (66)	62–70	433 (86.6)	83–90
Do you know what a self-test is?				
Yes	475 (95)	93–97	480 (96)	94–97
No	25 (5)	3–7	20 (4)	3–6
Do you perform self-test twice daily?				
Yes	495 (99)	97–99	475 (95)	93–97
No	5 (1)	0.3–2	25 (5)	3–7
Have you done another test for COVID-19?				
Yes	360 (72)	68–76	388 (77.6)	74–81
No	140 (28)	24–32	112 (22.4)	19–26
If yes which one?				
PCR	148 (41.1)	36–46	163 (42)	37–47
RADT	212 (58.9)	53–64	225 (58)	53–63
Did you feel anxiety or fear during the procedure of self-test?				
Yes	342 (69.1)	65–73	346 (72.9)	68–76
No	153 (30.9)	27–35	129 (27.1)	23–31
If yes how much?				
Very much	99 (28.9)	24–34	73 (21)	17–26
Much	137 (40.1)	35–45	133 (38.5)	33–44
Some	106 (31)	26–36	140 (40.5)	35–46
Does anyone of your friends/family have made a COVID test?				
Yes	424 (84.8)	81–88	430 (86)	82–89
No	76 (15.2)	12–19	70 (14)	11–17
Did you do the self-test by yourself?				
Yes	89 (18)	15–22	120 (25.2)	21–29
No with help	406 (82)	78–85	355 (74.8)	70–78
Did you face any problem during the procedure of self-test?				
Yes	342 (69.1)	65–73	351 (73.9)	70–78
No	153 (30.9)	27–35	124 (26.1)	22–30
If yes in which part of the procedure?				
Taking the sample	110 (32.2)	27–37	116 (33.1)	28–38
Putting the drops	34 (9.9)	7–13	33 (9.1)	6–13
Waiting the result	13 (3.8)	2–6	12 (3.4)	2–6
In the online government’s platform	185 (54.1)	49–59	190 (54.3)	49–60
Do you think you completed the procedure of self-testing correctly?				
Yes	366 (73.9)	69–77	375 (78.9)	75–82
No	129 (26.1)	22–30	100 (21.1)	17–25
Did you confirm the result of self-test with another method?				
Yes	24 (4.8)	3–7	38 (8)	6–11
No	471 (95.2)	93–97	437 (92)	89–94
If yes which method?				
PCR	5 (20.8)	8–42	7 (18.4)	8–34
RADT	19 (79.2)	57–92	31 (81.6)	65–92
Do you throw a negative self-test in the trash?				
Yes	360 (72)	68–76	385 (77.1)	73–81
No	140 (28)	24–32	115 (22.9)	19–27
If the self-test is positive do we throw it in the trash, inside 2 bags?				
Yes	370 (74)	70–78	365 (76)	72–80
No	130 (26)	22–30	115 (24)	20–28
Do we throw away the self-test in the recycle bin or in the drainage?				
Yes	240 (48)	43–52	206 (42.9)	38–47
No	260 (52)	47–56	274 (57.1)	52–61
Are you vaccinated against COVID-19?				
Yes	155 (31)	27–35	205 (41)	37–45
No	345 (69)	65–73	295 (59)	54–63
Do you afraid the side effects of the vaccine?				
Yes	116 (74.8)	67–80	137 (66.8)	60–73
No	39 (25.2)	19–33	68 (33.2)	27–40
If yes, how much?				
Very much	21 (18.1)	12–26	0 (0)	
Much	16 (13.8)	8–22	0 (0)	
Some	79 (68.1)	59–76	137 (100)	97–100
Which is your average school grade?				
<9.5 (E)	0 (0)		0 (0)	
9.5–13 (D)	40 (8)	6–11	45 (9)	7–12
13.1–16 (C)	360 (72)	68–76	335 (67)	63–71
16.1–18 (B)	75 (15)	12–18	90 (18)	15–22
18.1–20 (A)	25 (5)	3–7	30 (6)	4–8
During the school year, does anyone used internet or his/her phone to send you messages or images to make you feel uncomfortable, or ashamed?				
Yes	140 (28)	24–32	160 (32)	28–36
No	360 (72)	68–76	340 (68)	64–72
If yes, how many times?				
1 time	99 (70.7)	62–78	114 (71.3)	63–78
2 times	17 (12.1)	7–19	19 (11.9)	7–20
3 times	17 (12.1)	7–19	19 (11.9)	7–20
4 times	3 (2.1)	1–7	6 (3.8)	1–8
5 times	4 (2.9)	1–8	2 (1.3)	0.2–5
6 or more times	0 (0)		0 (0)	
During the school year, did you used internet or your phone to send messages or images to make feel uncomfortable, ashamed or harm any of your classmates?				
Yes	126 (25.2)	21–30	155 (31)	27–35
No	374 (74.8)	71–78	345 (69)	65–73
If yes, how many times?				
1 time	102 (81)	73–87	126 (81.3)	74–87
2 times	14 (11.1)	6.4–18	15 (9.7)	8–16
3 times	6 (4.8)	2–11	13 (8.4)	5–14
4 times	3 (2.4)	1–7	1 (0.6)	0.03–4.1
5 times	1 (0.8)	0.04–5.3	0 (0)	
6 or more times	0 (0)		0 (0)	

Abbreviations: PCR, Polymerase Chain Reaction test; RADT, Rapid Antigen Detection Test; CI, Confidence Interval; LL, Lower Limit of the confidence interval; UL, Upper Limit of the confidence interval.

**Table 2 ijerph-19-04559-t002:** Univariate analysis of COVID 19 self-test practice for students aged 13–15 y.o.

Variable	Do You Perform the Self-Test Twice by Week?		
	Yes N (%)	No N (%)	*p* Value
Sex			
Male	258 (52.1)	3 (60)	0.726
Female	237 (47.9)	2 (40)	
COVID-19 knowledge			
Yes	483 (97.6)	5 (100)	0.725
No	12 (2.4)	0 (0)	
Have you been infected by COVID-19?			
Yes	80 (16.2)	0 (0)	0.327
No	415 (83.8)	5 (100)	
Has anyone of your family environment been infected by COVID-19?			
Yes	169 (34.1)	1 (20)	0.507
No	326 (65.9)	4 (80)	
Are you vaccinated against COVID-19?			
Yes	154 (31.1)	1 (20)	0.593
No	341 (68.9)	4 (80)	
Which is your average school grade?			
16.1–18/18.1–20 (A, B)	99 (20)	1 (20)	1.000
9.5–13/13.1–16 (C, D)	396 (80)	4 (80)	
During the school year, does anyone used internet or his/her phone to send you messages or images to make you feel uncomfortable, or ashamed?			
Yes	138 (27.9)	2 (40)	0.548
No	357 (72.1)	3 (60)	
During the school year, did you used internet or your phone to send messages or images to make feel uncomfortable, ashamed or harm any of your classmates?			
Yes	126 (25.5)	0 (0)	0.192
No	369 (74.5)	5 (100)	

**Table 3 ijerph-19-04559-t003:** Univariate analysis of COVID 19 self-test practice for students aged 16–18 y.o.

Variable	Do You Perform the Self-Test Twice by Week?		
	Yes N (%)	No N (%)	*p* Value
Sex			
Male	228 (48)	3 (60)	0.593
Female	247 (52)	2 (40)	
COVID-19 knowledge			
Yes	473 (99.6)	5 (100)	0.884
No	2 (0.4)	0 (0)	
Have you been infected by COVID-19?			
Yes	52 (10.9)	2 (40)	0.041
No	423 (89.1)	3 (60)	
Has anyone of your family environment been infected by COVID-19?			
Yes	65 (13.7)	0 (0)	0.374
No	410 (86.3)	5 (100)	
Are you vaccinated against COVID-19?			
Yes	195 (41.1)	2 (40)	0.962
No	280 (58.9)	3 (60)	
Which is your average school grade?			
16.1–18/18.1–20 (A, B)	115 (24.2)	1 (20)	0.827
9.5–13/13.1–16 (C, D)	360 (75.8)	4 (80)	
During the school year, does anyone used internet or his/her phone to send you messages or images to make you feel uncomfortable, or ashamed?			
Yes	155 (32.6)	1 (20)	0.549
No	320 (67.5)	4 (80)	
During the school year, did you used internet or your phone to send messages or images to make feel uncomfortable, ashamed or harm any of your classmates?			
Yes	146 (30.7)	2 (40)	0.655
No	329 (69.3)	3 (60)	

**Table 4 ijerph-19-04559-t004:** Univariate analysis of performing the self-test for both age categories.

Variable	Do You Perform the Self-Test Twice by Week?		
	Yes N (%)	No N (%)	*p* Value
Age category			
13–15 y.o.	495 (51)	5 (16.7)	<0.001
16–18 y.o.	475 (49)	25 (83.3)	
Sex			
Male	486 (50.1)	17 (56.7)	0.479
Female	484 (49.9)	13 (43.3)	
COVID-19 knowledge			
Yes	956 (98.6)	30 (100)	0.508
No	14 (1.4)	0 (0)	
Have you been infected by COVID-19?			
Yes	132 (13.6)	3 (10)	0.569
No	838 (86.4)	27 (90)	
Has anyone of your family environment been infected by COVID-19?			
Yes	234 (24.1)	3 (10)	0.073
No	736 (75.9)	27 (90)	
Are you vaccinated against COVID-19?			
Yes	349 (36)	11 (36.7)	0.938
No	621 (64)	19 (63.3)	
Which is your average school grade?			
16.1–18/18.1–20 (A, B)	214 (22.1)	6 (20)	0.788
9.5–13/13.1–16 (C, D)	756 (77.9)	24 (80)	
During the school year, does anyone used internet or his/her phone to send you messages or images to make you feel uncomfortable, or ashamed?			
Yes	293 (30.2)	7 (23.3)	0.418
No	677 (69.8)	23 (76.7)	
During the school year, did you used internet or your phone to send messages or images to make feel uncomfortable, ashamed or harm any of your classmates?			
Yes	272 (28)	9 (30)	0.814
No	698 (72)	21 (70)	

## Data Availability

The study data are available from the corresponding author upon reasonable request.

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
