# Peer review of "Evaluation of Knowledge, Attitudes and Practices Related to Self-Testing Procedure against COVID-19 among Greek Students: A Pilot Study"

_ijerph, 2022, doi:10.3390/ijerph19084559_

Round 1

Reviewer 1 Report

Thank you for giving me the opportunity to read and comment a report “The self-testing procedure of Greek studentes against Covid-19, a way to reduce the spread and control the pandemic?”, by Marinos G, et al.

This is a potentially interesting report but at present it is not suitable for publication

Major revision:

  • It would be desirable for the authors to indicate the main aim of the study, since it is not mentioned in the manuscript.
  • It would be appropriate that the categorical variables represented in terms of frequency distribution are accompanied by their corresponding confidence intervals.
  • The authors indicate that they will use the Chi-square test to determine differences between subgroups. Perhaps it would be appropriate to add the p results in Table 1 for each subgroup.
  • In the text of the results section, it is not necessary to include the value of N for each percentage, since it is reiterative. This reviewer recommends including the N values only in Table 1.
  • It is not necessary to include a subsection in the results section to include the table. It can simply be mentioned in the text and added below.
  • This reviewer recommends modifying Table 1. First, modify the title to a more descriptive one, second, include the p values for the comparison between subgroups and, third, revise the structure of the table since there are disordered values.
  • Conclusions must be derived from the results obtained in the investigation. Therefore, it is recommended to review the first paragraph of the section.

Minor revision:

  • The structure of the abstract does not follow the recommendations of the journal. The abstract should be a single paragraph and should follow the style of structured abstracts, but without headings.
  • The acronym COVID-19 is written in different ways throughout the report, for example, Covid-19 or COVID-19. Please unify the terms.
  • The format of some references is incorrect.

Author Response

Dear author,

Thank you for your comments. All were well accepted.

The aim of the study is added.

Table 1 is formatted and the requested data are added.

Three new tables are added , recording the univariate analysis and p-value for each age category and for the whole sample of students.

Conclusion has been rephrased.

The abstract is now following the recommendations of the journal.

The acronyms and the references were checked.

Reviewer 2 Report

The manuscript analyses self-test awareness among Greek students from the Attika region through a questionnaire to students aged between 13 and 18 years out of a total of 1000 students. The introduction does not present the objective of the study, making only a contextualization of the procedures related to the opening of schools after the lockdown period by the Greek Government to control local outbreaks of COVID-19. The questionnaire includes questions about knowing the self-test, conducting self-test, average grade and bullying but there are not any correlation analyses and the results doesn’t show that self-test procedure reduce the spread and control the pandemic. The results could be presented in the form of graphs to be easily identified. The results are not correlated with the title of the manuscript and it should be changed. The study needs to be reformulated.

Author Response

Dear author.

Thank you for your comments. All are well accepted.

The aim of the study was added. 

The title is now changed, so the results to be correlated.

Three new tables were added , recording the univariate analysis and p-value in each age group and for the whole sample of students.

The Greek government in an effort to reduce the spread of COVID-19 during the re-opening of schools has implemented a rule that each student has to perform a RADT with the procedure of self-testing , so infected teenagers ,with or without symptoms will be early identified , quickly isolated ,not attending school lessons and as a result not spreading COVID-19 in a closed community like school and therefore to their family and friends.  As described in the manuscript the aim of this study was to evaluate the knowledge of self testing among Greek students and  all the factors that are associated with this procedure.

We hope that , after your proposals and after the configuration of the manuscript we meet most of your expectations.

Round 2

Reviewer 1 Report

The manuscript has not improved enough to be considered for publication.

The authors have written the introduction section in a single paragraph. This is not usual, and it is recommended to restructure the introduction section, using a paragraph for each main topic.

Confidence intervals have not been described in the statistical analysis section

In the results section there are many values with punctuation errors. Decimals are indicated by points, not commas.

I honestly do not understand the confidence intervals shown in Table 1. Why are they not calculated for all questions? Also, in the foot of Table 1, the corresponding abbreviations should be included.

Different terms are still shown for COVID-19. For example, COVID-19 (line 48) and COVID (line 97). It would be necessary to unify the terms.

As indicated in the first review, the first paragraph of the conclusions is not based on the authors' results. It would need to be reworded or removed.

In conclusion, the manuscript needs to be reformulated.

Reviewer 2 Report

The manuscript has provided some additional information and it was improved but still missing some graphs for the results to be easily identified. The list of tables without any explanation doesn’t seem the best way to help the reader.

Round 3

Reviewer 1 Report

The manuscript has improves and us now suitable for publication.

However, I would just like to suggest a minor revision. The confidence intervals in Table 1 should be multiplied by one hundred to express them as a percentage.

Author Response

Thank you for giving me the opportunity to submit a revised draft of our manuscript titled “Evaluation of knowledge, attitudes and practices related to self-testing procedure against COVID-19 among Greek students: a pilot study”, to International Journal of Environmental Research and Public Health.  We appreciate the time and effort that you and the reviewers have dedicated to providing your valuable feedback on our manuscript.  We have highlighted the changes within the manuscript.

Point 1: However, I would just like to suggest a minor revision. The confidence intervals in Table 1 should be multiplied by one hundred to express them as a percentage

Response 1: Thank you for pointing this out. We agree with this comment. Therefore, we have modified the confidence intervals in Table 1.

We thank the reviewer for his/her thoughtful and thorough review and believe his/her input has been invaluable to make our manuscript more balanced.

Sincerely,

Lamprinos Dimitrios

Reviewer 2 Report

 The manuscript has been improved in order to be published

Author Response

 We thank the reviewer for his/her thoughtful and thorough review and believe his/her input has been invaluable to make our manuscript more balanced.

Sincerely,

Lamprinos Dimitrios